# The Use of Vibration Training in Men after Myocardial Infarction

**DOI:** 10.3390/ijerph19063326

**Published:** 2022-03-11

**Authors:** Agata Nowak-Lis, Zbigniew Nowak, Tomasz Gabrys, Urszula Szmatlan-Gabrys, Ladislav Batalik, Vera Knappova

**Affiliations:** 1Department of Physiotherapy, Jerzy Kukuczka’s Academy of Physical Education, 40-065 Katowice, Poland; zbinow@gmail.com; 2Department of Physical Education and Sport Science, Faculty of Pedagogy, University of West Bohemia, 30100 Pilsen, Czech Republic; tomaszek1960@o2.pl (T.G.); knappova@ktv.zcu.cz (V.K.); 3Department Anathomy, Faculty of Rehabilitation, University of Physical Education, 31-571 Krakow, Poland; ulagabrys1957@tlen.pl; 4Department of Rehabilitation, University Hospital Brno, 62500 Brno, Czech Republic; batalik.ladislav@fnbrno.cz; 5Department of Public Health, Faculty of Medicine, Masaryk University, 62500 Brno, Czech Republic

**Keywords:** comprehensive cardiac rehabilitation, vibration platform, percutaneous coronary angioplasty, ischemic heart disease, myocardial infarction

## Abstract

The aim of the study was to evaluate the effects of the applied whole-body vibration training (WBV) as additional training to standard rehabilitation programme on exercise tolerance, evaluated through an exercise test, blood lipid profile, and the changes in selected echocardiographic parameters of patients after myocardial infarction. The study involved 63 males. The subjects were divided into two groups: standard—ST (27) and with vibration training—ST + WBV (36). All the subjects had undergone angioplasty with stent implantation. The standard and with vibration training group carried out a 24-day improvement program comprising 22 training units. Each session consisted of endurance, general stamina, and resistance training. Instead of resistance training, the experimental group performed exercises on the vibration platform. Statistically significant changes in both groups were observed in the parameters of the echocardiographic exercise test, such as test duration (*p* < 0.001), distance covered (*p* < 0.001), MET (*p* < 0.001), VO_2_max (*p* < 0.001), and HRrest (*p* < 0.01). The echocardiographic test revealed significant improvement of Left Ventricular Ejection Fraction in both groups (ST + WBV group *p* = 0.024, ST group *p* = 0.005). There were no statistically significant changes in blood lipid profile and body mass and composition.

## 1. Introduction

Guidelines for secondary prevention interventions indicate multiple strategies for cardiovascular risk control [1]. Properly controlled physical activity and exercise training are recommended in all patients with cardiovascular diseases. It is known that training is used as the gold standard of exercise-based cardiac rehabilitation (ex-CR) [2] and can lead to a significant impact on quality of life, morbidity and mortality [3]. It is rare to find publications describing the practical application of modern, innovative forms of ex-CR [4,5] However, innovative training forms require adaptation to the needs of cardiac patients. One such form is whole-body vibration training (WBV). There are many applications of vibrotherapy in medical rehabilitation. It can affect the body comprehensively by improving blood circulation [6,7] and lymph [8], stimulating the neuromuscular system, including deep sensation [9,10,11]. It can also be used in pain treatment [12,13] or spasticity [14]. It has also found application in gyms, fitness clubs and sports training [15,16]. The use of therapeutic vibrations, due to obtaining similar effects, is compared to moderate-intensity exercise [17,18]. Moreover, the main benefits of WBV have been observed in positive effects on the neuromuscular system [15], bone density [19] and cognitive function [20]. Changes in lean mass are a frequent critical determinant in the pathophysiology of heart failure, and sarcopenia may be considered one of the leading reasons for low physical performance and decreased cardiorespiratory fitness in elderly patients with chronic diseases [21]. The prevalence of sarcopenia in chronic heart failure patients is up to 20% and may progress in cardiac cachexia. Muscle wasting is a strong predictor of frailty and decreased survival [22], thus it is essential to develop preventive strategies to influence this condition.

There is still little information on the possibility of using this form of training in the group of patients with ischemic disease or myocardial infarction. Vibrotherapy as an alternative method of ex-CR is not widely used yet, but some authors point to the great potential of this method [23].

The use of a vibration platform may turn out to be a good alternative to the sometimes unattractive exercises that have been used thus far during outpatient rehabilitation. The aim of this research was to evaluate the effects of the applied whole-body vibration training (WBV) as a modern therapeutic strategy in patients diagnosed with ischemic disease or myocardial infarction.

The following research questions were asked:May the level of exercise tolerance assessed with an electrocardiographic test exercise stress be related to the type of training used (standard vs. vibration training)?Can the applied training forms affect the hemodynamic parameters of the left ventricle (assessed by echocardiography), profile lipid (laboratory analysis), and body weight composition (evaluated method bioimpedance)?

## 2. Materials and Methods

### 2.1. Characteristics of the Tested Material

The study involved 63 males classified as NYHA I. The patients were randomly assigned (random selection by drawing cards with the name of the group) based on 2 therapeutic strategies:

Standard group (ST group)—27 people aged 55.19 ± 8.03 years. Patients subjected to improvement based on the recommendations of the European Society of Cardiology. Detailed characteristics of the study groups are included in Table 1 and Table 2. 

Group with vibration training (ST + WBV group)—36 people aged 53.71 ± 7.13 years, in whom whole body vibration training (WBV) was used additionally. Descriptive characteristics of the subjects.

In both analysed groups, patients with angioplasty and one implanted stent represented the highest percentage.

The research was approved by the Bioethics Committee for Scientific Research at the Jerzy Kukuczka Academy of Physical Education in Katowice (No. 4/2010). All patients were informed and gave written consent to participate in the project. Anyone could discontinue participation at any time. None of the patients changed their physical activity or diet during the studies. Pharmacological treatment was also not changed. Patients came to rehabilitation alone or in a car. 

Inclusion criteria:NYHA I
documented ischemic heart disease or myocardial infarction,Time since the infarction not less than 2 months and not more than 6 monthsPatients with good exercise tolerance assessed by exercise test: ≥7 MET,


Exclusion criteria:Unregulated hypertensionUnstable anginaRecent myocardial infarction <2 months after the incidentArrhythmias and conduction disturbancesVaricose veins of the lower extremitiesPrevious unhealed lower limb injuriesAdvanced peripheral arteriosclerosisDiagnosed neoplastic diseaseDiseases of the central or peripheral nervous systemEpilepsyEF% ≤ 35Age ≥ 75

All patients took their medications as prescribed by their treating physicians. None of the patients had dose adjustments.

In Figure 1 there is shown a randomization process.

### 2.2. Experimental Procedure

Patients qualified for the study underwent a rehabilitation program, including 22 training units performed 5 times a week, in accordance with the adopted ESC standards. In the ST + WBV group, patients started with a 10-min warm-up on a bicycle ergometer (minimum load 25 WAT), and then were trained on a vibration platform for 20 min. The total time of exposure to the vibration was 10 min because the 1:1 interval training was used. Patients maintained a half-squat for 60 s. The bending angle of the knee and hip joints was 40°, the heels detached from the ground, and the hands were held on the platform handle (Figure 2). Each person exercised without shoes and at the same time of the day. During the break, the patients rested in a sitting position. The platform generated vibrations with a frequency of 40 Hz and an amplitude of 2 mm.

The ST + WBV group performed exercises for the upper and lower body on a vibration platform (Powerplate, Amsterdam, The Netherlands).

The following procedure was carried out before commencing the training program and immediately after its completion: electrocardiographic submaximal exercise test on a treadmill (6-stage Bruce protocol: stage 1 = 2.7 km/h, 10%, stage 2 = 4.0 km/h, 12%, stage 3 = 5.5 km/h, 14%, stage 4 = 6.8 km/h, 16%, stage 5 = 8.0 km/h, 18%, stage 6 = 8.8 km/h, 20%). The following physiological variables were measured: test duration (min), distance covered (m), energy cost (MET), heart rate at rest (HRrest; 1/min) and maximum (HRmax; 1/min), blood pressure at rest and maximum systolic (BPSrest, BPSmax; mmHg) and diastolic (BPDrest, BPDmax; mmHg), and peak oxygen consumption (VO_2_peak) per kilogram of bodyweight. Spiroergometric parameters were determined with the CORTEX portable METAMAX 3B gas analyser exercise test using the Excalibur Sport cycle ergometer (Lode, Groningen, The Netherlands).

Two-dimensional ultrasound heart test, measured parameters: LVEDD—left ventricular end-diastolic dimension, LVESD—left ventricular end-systolic dimension, LVESV—left ventricular end-systolic volume, LVEDV—left ventricular end-diastolic volume LVSV—left ventricular stroke volume, LVEF—left ventricular ejection fraction, LVM—left ventricular mass, LVMI—left ventricular mass index. Echocardiography was performed by the ALT HDI 3000 ultrasound (Philips: Bothell, United States.). 

Blood lipid profile test. Measured parameters: TC—total cholesterol, HDL—high-density lipoproteins, LDL—low-density lipoproteins, TG—triglycerides 

The study was carried out in an analytical laboratory by a qualified person. 

Body weight composition (bioimpedance method, using Tanita TBF-300 scale). Parameters taken into account: BMI—body mass index, FP (%)—fat percent, FM—fat mass, FFM—fat-free body mass, TBW—total body water, BMR—basal metabolic rate.

### 2.3. Data Analysis

The Excel 2007 Microsoft spreadsheet was used to develop the clinical material database. Additional variables were also calculated in this program. The database was implemented into the licensed Statistica PL package by StatSoft.

In the first stage, the basic descriptive statistics of the collected data of the interval scale were determined (mean, variance, standard deviation, standard error of the mean, median, fashion, quantiles, extreme values). For each parameter, the Shapiro–Wilk test, Kolmogorov–Smirnov test, and Lilliefors normality test was carried out in order to verify the hypothesis about the consistency of the distribution of the examined feature with the normal distribution, and the sample randomness test with the series test. The level of significance was *p* (α) = 0.05.

For variables with a distribution close to normal, the following were used:-Parametric Student’s t-test for independent variables, preceded by Fisher’s test of homogeneity of variance. In the case of lack of homogeneity of variance, the Satterwhite test was used-Parametric test for differences (related features)—Student’s *t*-test-One-way analysis of variance preceded by Bartlett’s test of homogeneity of variance.

For distributions deviating from the normal distribution, the following were used: non-parametric Wilcoxon test, Kruskal–Wallis ANOVA test, Spearman’s rank correlation test.

## 3. Results

### 3.1. Electrocardiographic Exercise Test

In both studied groups, there was a significant change in the level of exercise tolerance in terms of test duration, distance covered, metabolic cost, as well as maximum oxygen consumption, and resting heart rate. However, no statistically significant differences (delta) were found in comparisons between the groups of changes in the assessed indicators (Table 3).

### 3.2. Echocardiographic Test

In both studied groups, statistically significant changes were found only in the left ventricular ejection fraction (LVEF). In the case of the remaining analyzed indicators, positive changes were obtained, but they were not statistically significant. (Table 4). 

### 3.3. Blood Lipid Profile 

In both analyzed groups, no significant changes in the blood lipid profile were observed. 

The intergroup assessment of changes (delta) also did not show statistically significant differences (Table 5).

### 3.4. Body Composition Testing

Both the ST + WBV and ST groups, no significant changes in the blood lipid profile were observed. Intergroup comparison of changes (delta) also showed no statistically significant differences (Table 6).

## 4. Discussion

In order to develop and maintain cardiovascular and respiratory fitness, it is necessary to maintain physical activity [24,25]. The study assessed and compared the effectiveness of two different training methods, the well-known and widely used standard method and the new one, vibrotherapy, which is likely to enter the rehabilitation program for patients after acute coronary syndrome (ACS). The obtained results confirmed the effectiveness of the standard method, which was expected even before the start of the study, but above all showed that vibration training can be included in the rehabilitation program and have similar effects.

### 4.1. Electrocardiographic Exercise Test

The results obtained after completion of the 24-day rehabilitation program in relation to the results obtained before its commencement indicate a significant improvement in exercise tolerance. In both analyzed groups, i.e., ST + WBV (in which vibration training was used) and the group training according to the adopted ESC standards (ST group), a statistically significant increase in the same indicators of exercise tolerance (test time, distance, MET, VO_2_max, HRrest) was observed. Indicators that are very often assessed during an exercise test and show the level of endurance are the test duration and the distance covered on the treadmill.

The obtained statistically significant increase in the duration of the exercise test in the ST + WBV and ST, as well as the increase in the distance covered, proves the high effectiveness of both forms of training. Other authors also obtained similar results [26,27,28]

Another indicator of exercise tolerance, assessed in the exercise ECG test, is the metabolic equivalent of MET. According to Myers et al. [29], peak exercise capacity measured in METs is a very good prognostic factor for the risk of death both in patients with cardiovascular diseases and in healthy people. Own research has shown that both training with the use of the vibration platform (WBV) and the standard method significantly improved the MET index. A favorable increase in the value of MET associated with the conducted cardiac rehabilitation program in the second stage was often demonstrated in relation to both the standard program [30] and the modified one [4,31]

Maximum oxygen consumption (VO_2_max) is the basic indicator determining endurance, especially during prolonged exercise, and at the same time assessing the capacity of the cardiovascular system [32]. Severe heart failure is assessed at the level of 10 mL/kg/min. The minimum level of physical fitness assessed by VO_2_max is 40 mL/kg/min. For a sedentary person, the VO_2_max value is approximately 30 mL/kg/min. [33]. Based on the results obtained, a significant increase in VO_2_max was observed in both training groups, WBV and ST, which suggests that a properly planned and implemented rehabilitation program conducted in a continuous and systematic manner leads to a significant improvement in the physical capacity of patients. In addition, Yang et al. [34], Guazzi et al. [35], and Adams et al. [36] reached similar conclusions in their research, thus corroborating the positive effect of cardiac rehabilitation on the spiroergometric indicators of physical fitness in patients with heart failure and after acute coronary events.

The effect of proper adaptation to physical exertion, which at the same time indicates an increase in exercise capacity, is a decrease of the resting and peak heart rate. It occurs by reducing the activity of the autonomic nervous system. However, according to some researchers, an increased resting heart rate may be an independent risk factor for cardiovascular events in both men and women.

In our own research, both in the WBV and ST groups, a decrease in the value of the resting heart rate was noted. A similar effect in relation to the standard program and modified programs was obtained by Grabara et al. [5] and Nowak et al. [31,37].

### 4.2. Echocardiographic Test

The indicators of the left ventricle, important for the effectiveness of the rehabilitation program, were assessed. The period of 22 days of training is quite short to expect significant changes in the hemodynamics of the left ventricle, which was confirmed by the results obtained in both training groups. Nevertheless, the substantial increase in LVEF shows an improvement in left ventricular contractility as a result of the training program as this increase was recorded in both groups, it should be assumed that both the training using the vibration platform (ST + WBV group) and the standard program (ST group) were equally effective. It can only be assumed that subsequent studies after the next 3 or 6 months would show significant differences in relation to the other parameters.

Left ventricular ejection fraction is an indicator of the global contractility of the heart muscle and is one of the most important parameters determining the condition of patients after myocardial infarction [28]. It is also a parameter reflecting the effectiveness of comprehensive cardiac rehabilitation, as evidenced by the studies by Doimo et al. [38].

Belardinelli et al. [39], in a 6-month follow-up, found that the LVEF value was an indicator that significantly differentiated patients. In active people, the authors observed a significant increase (52.3 vs. 57.3%, *p* < 0.000), which is consistent with the changes observed in our own analysis. Fahreen et al. [40], after completing a 6-week rehabilitation program in people after a myocardial infarction, found a significant improvement in the value of left ventricular ejection fraction in the group of combined resistance and aerobic training (45 vs. 55%, respectively; *p* = 0.029 and 45 vs. 50%; ns).

The influence of physical training used in the second stage of rehabilitation on the work of the heart muscle has not been clearly explained. In most studies, as well as in our own studies, it was not possible to demonstrate a significant effect of training on the morphological and functional parameters of the left ventricle, except, of course, for the LVEF parameter. It should be noted that both forms of the applied training, both ST and WBV influenced its (LVEF) growth, which is considered by many researchers to be one of the most important prognostic parameters in patients after myocardial infarction [41].

### 4.3. Blood Lipid Profile

Increased levels of total cholesterol and triglycerides are factors in the formation of atherosclerotic lesions in the coronary, cerebral, and peripheral vessels. Their concentrations in blood serum are determined hereditarily, but a significant role in lowering the levels is attributed to lifestyle elements (environmental factors), such as a proper diet and systematic physical activity [42,43]. Scientific reports confirm the beneficial effect of physical activity on the lipid profile, although it concerns longer observations, e.g., 6 months [44]. In the case of observations that cover a short period of time, the changes are not statistically significant, which was also the case in our own research. It is also difficult to say whether the reason for the changes observed is the rehabilitation program or the effect of statins. Comparing the results of the tests before and after the start of rehabilitation, the level of the analyzed lipids in both cases was within the normal range, which may be even more indicative of the earlier administration of pharmacological treatment.

### 4.4. Body Composition Testing

Body mass index (BMI) analysis is most widely used in assessing body weight. Its reduction is recommended for people with a BMI greater than 24.9 kg/m^2^ and for women and men whose waist circumference exceeds 88 and 102 cm, respectively. According to the guidelines of the European Society of Cardiology (ESC), these values are accepted as the norm [45]. Ashton et al., in a study of 14,077 women aged 30–64 years, observed a significant increase in the risk of coronary heart disease from a BMI of 22 kg/m^2^ [46]. However, the usefulness of this variable is increasingly being questioned [45]. Therefore, in the present study, an additional analysis of the patient’s body weight was performed using the bioimpedance method. There are, however, factors that may affect the accuracy of measurements such as exercise, alcohol consumption, diuretic medication, edema, and, most likely to a minor extent, the menstrual cycle. Due to the fact that exercise can also affect the distribution of adipose tissue in the body, reducing the so-called visceral obesity, the lack of waist circumference measurement may be a limitation of this study. This index is used to assess abdominal obesity associated with the risk of hypertension, hyperglycemia, and disturbances in the lipid profile. These results would also allow for comparison with the variables obtained by the bioimpedance method.

Vissers et al. [47] observed significant, positive changes in the composition of body weight under the influence of vibrations used in training. This was also confirmed in the research by Artero et al. [48], who reported that supplementing resistance training with vibration training reduces body fat. With traditional exercises, this impact was minimal. Perhaps the intensity of exercises and the duration of the training cycle used in the research turned out to be the reason for the statistical significance of the obtained results. However, it is worth emphasizing the positive direction of changes in body mass composition indices. Similar results were obtained in the studies by Jarska et al. [49]. More and more often in scientific reports, there is an opinion that the best benefits of physical activity as a therapeutic intervention in weight reduction can be obtained in conjunction with an appropriately selected diet [50,51].

### 4.5. Summary

Vibrotherapy, as an innovative method of cardiac rehabilitation, can be an excellent alternative, especially for people who are weak and unable to exercise intensively after cardiovascular diseases. It can also be assumed that obese patients who are unable to perform certain exercises will benefit similarly from low-frequency vibration training. In addition, patients with circulatory failure in NYHA class II/III (and possibly class III) will be able to improve their physical capacity after several 20-min sessions on a vibration platform. The very fact that there are no differences in the end result of both forms of training is already promising.

### 4.6. Limitations

In order to unequivocally confirm the usefulness of vibration training in a cardiac rehabilitation program, such studies should be carried out on a much larger group of patients, not only men but also women. Qualification for the study should include not only patients in NYHA class I, but also patients in NYHA classes II and III due to the lack of differences in the final effect of both forms of training.

## 5. Conclusions

In both study groups (ST+ WBV and ST), a significant improvement in exercise tolerance was achieved, as assessed on the basis of the exercise electrocardiographic test.The applied training forms caused only a significant improvement in the left ventricular ejection fraction (LVEF) but did not change the lipid profile or body weight composition.Obtaining a similar training effect and the lack of statistically significant differences in intergroup comparisons confirm the usefulness of vibrotherapy in cardiac rehabilitation.

## Figures and Tables

**Figure 1 ijerph-19-03326-f001:**
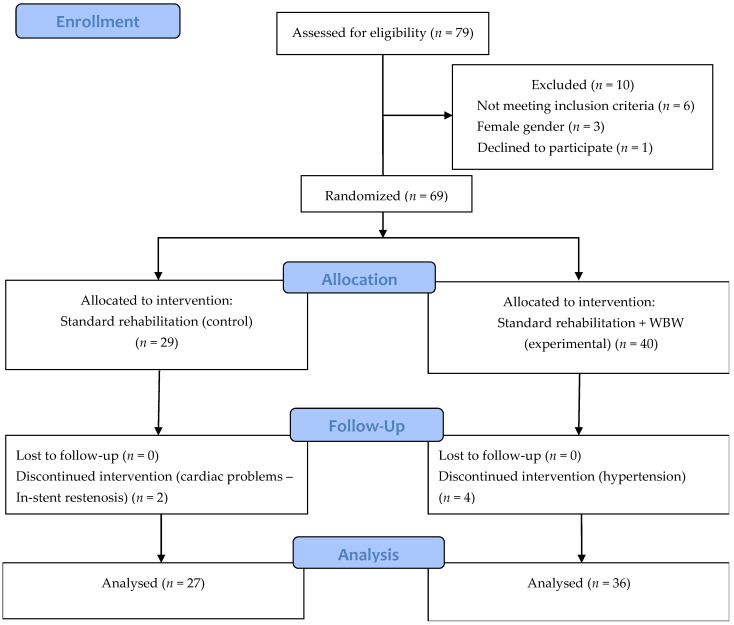
Flow diagram. * Too small number that could change results of statistical analysis.

**Figure 2 ijerph-19-03326-f002:**
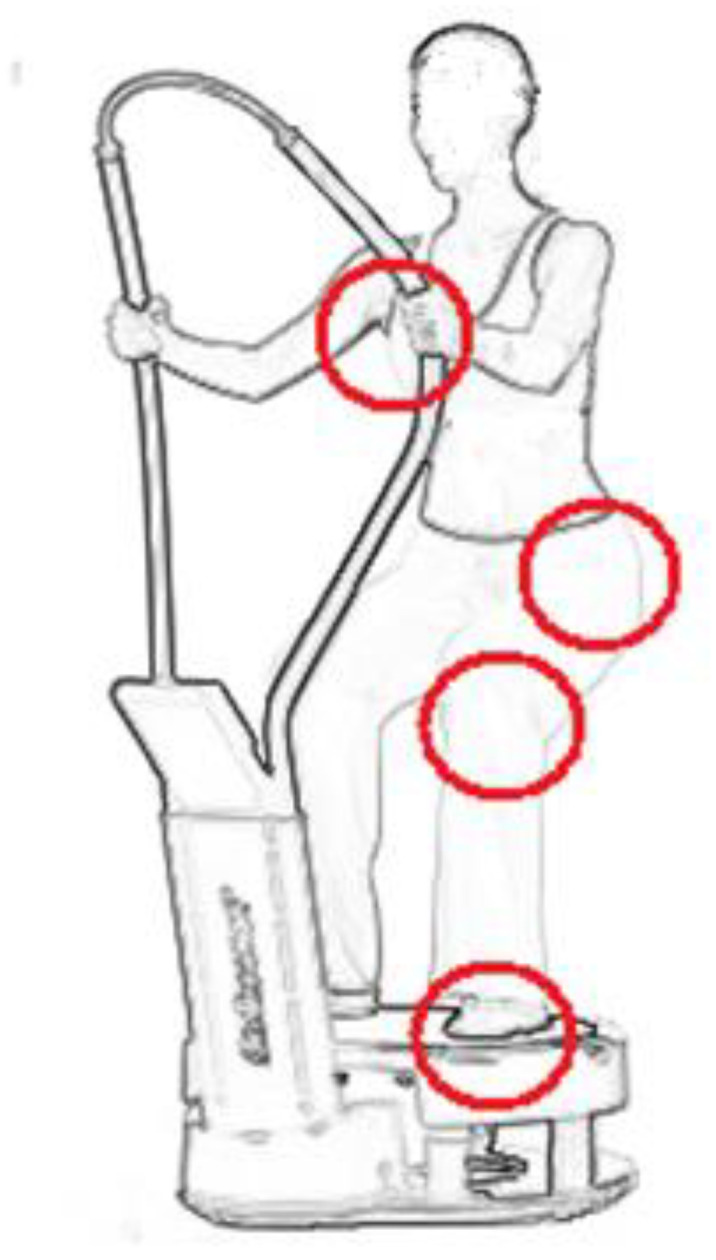
Body position during the exercise.

**Table 1 ijerph-19-03326-t001:** Training following ESC recommendations.

Training Type	Methodology	Load
Endurance training	Training on a stationary bicycle, 5 times a week for 30 min	The load applied on the basis of the calculated training heart rate starting with 60% of the heart rate reserve and increasing it by10% after 5 training units, up to 80% of the heart rate reserve, up to 14th degree of the Borg scale for perceived exertion
General stamina training	Gymnasium exercises—elements of aerobic and anaerobic training, stretching, breathing exercises	
	- general stamina training in a gym with steppers, gym balls, mattresses, and wooden sticks (150 cm), - 5 times a week for 30 min	
Resistance training	Resistance training performed for 30 min in a strength training room using elliptical trainers, rowing machines and steppers 5 times a week for 30 min.	

**Table 2 ijerph-19-03326-t002:** Characteristics of study groups.

Condition Type	ST+ WBV Group	ST Group
	N(%)	N(%)
Age [years]	53.71 ± 7.13	55.19 ± 8.03
Types of coexisting conditions		
Ischemic heart disease	29 (80.5%)	22 (81.4%)
Type 2 diabetes	4 (11.1%)	2 (7.4%)
Hyperlipidemia	31 (86.1%)	21 (77.7%)
Hypertension	8 (22.2%)	5 (18.5%)
Myocardial infarction	36 (100%)	27 (100%)
NSTEMI	20 (55.5%)	13 (48.1%)
STEMI	16 (44.5%)	14 (51.9%)
Applied treatment type		
PTCA + STENT	36 (100%)	27 (100%)
PTCA	0	0
1 stent	28 (77.8%)	16 (59.3%)
2 stents	8 (22.2%)	10 (37%)
3 stents	0	1 (3.7%)
≥4 stenst	0	0

**Table 3 ijerph-19-03326-t003:** CPET results.

Variable	ST+ WBV Group	*p*	ST Group	*p*	Δ ST+ WBV vs. ΔST
Time I	8.52 ± 1.15	<0.001	7.57 ± 1.10	<0.001	0.155
Time II	9.74 ± 1.35		9.56 ± 1.76		
Δ [min]	1.2		2		
Distance I	356.76 ± 72.04	<0.01	321.73 ± 62.48	<0.001	0.192
Distance II	433.18 ± 87.50		424.15 ± 80.25		
Δ [m]	76.4		102.42		
MET I	10.95 ± 1.19	<0.001	9.99 ± 1.08	<0.001	0.392
MET II	12.05 ± 1.50		11.91 ± 1.86		
Δ	1.03		1.92		
VO_2_max I	38.59 ± 5.91	<0.001	34.21 ± 5.07	<0.001	0.175
VO_2_max II	45.17 ± 7.45		44.53 ± 9.25		
Δ [mL/kg/min]	6.60		10.33		
HRrest I	68.65 ± 8.47	0.032	70.10 ± 8.08	0.012	0.997
HRrest II	64.94 ± 9.31		66.25 ± 8.55		
Δ [bpm]	−3.7		−3.85		
HRmax I	130.24 ± 11.92	0.122	122.35 ± 15.16	0.379	0.802
HRmax II	135.41 ± 15.16		124.65 ± 13.54		
Δ [bpm]	5.2		2.30		
SBPrest I	125.00 ± 13.22	0.332	124.75 ± 14.00	0.425	0.596
SBPrest II	122.65 ± 10.32		123.50 ± 6.39		
Δ [mmHg]	−2.4		−1.25		
DBPrest I	80.59 ± 4.49	0.058	78.75 ± 6.46	0.449	0.070
DBPrest II	79.41 ± 4.29		77.50 ± 6.39		
Δ [mmHg]	−1.18		−1.25		
SBPmax I	155.00 ± 14.36	0.058	150.25 ± 15.93	0.292	0.845
SBPmax II	148.82 ± 14.95		146.50 ± 10.89		
Δ [mmHg]	−6.18		−3.75		
DBPmax I	85.29 ± 6.24	0.075	81.25 ± 7.23	0.489	0.863
DBPmax II	81.76 ± 76.36		80.00 ± 4.59		
Δ [mmHg]	−3.53		−1.25		

All data are presented as mean values ± standard deviations and difference (Δ—delta), *p*—statistically significant level (the lowest level was *p* ≤ 0.05), MET—metabolic equivalent, VO_2_max—maximal oxygen uptake, HRrest—resting heart rate, HRmax—heart rate maximal, SBPrest—resting systolic blood pressure, DBPrest—resting diastolic blood pressure, DBPmax—maximum diastolic blood pressure.

**Table 4 ijerph-19-03326-t004:** Echocardiographic test results in both study groups.

Variable	ST+ WBV Group	*p*	ST Group	*p*	Δ ST+ WBV vs. Δ ST
LVEDD I	50.47 ± 5.05	0.345	50.35 ± 5.11	0.724	0.718
LVEDD II	49.58 ± 4.63		50.05 ± 4.85		
Δ [mm]	−0.89		−0.30		
LVESD I	34 ± 5.20	0.221	33.90 ± 3.88	0.183	0.441
LVESD II	33.05 ± 5.60		34.65 ± 4.26		
Δ [mm]	0.941		0.75		
LVESV I	49.08 ± 16.03	0.287	48.03 ± 13.21	0.157	0.877
LVESV II	46.22 ± 17.10		50.78 ± 15.19		
Δ [mL]	−2.854		2.75		
LVEDV I	122.46 ± 27.81	0.431	121.85 ± 28.53	0.530	0.748
LVEDV II	121.44 ± 24.80		120.03 ± 27.00		
Δ [mL]	−1.02		−1.82		
LVSV I	91.38 ± 26.45	0.188	90.98 ± 32.36	0.250	0.521
LVSV II	92.35 ± 22.01		91.33 ± 28.97		
Δ [mL]	−1.503		−2.65		
LVEF I	53.18 ± 3.73	0.024	53.30 ± 3.06	0.005	0.321
LVEF II	54.53 ± 4.06		55.30 ± 4.13		
Δ [%]	1.35		2.00		
LVM I	191 ± 43.98	0.112	193.88 ± 39.82	0.177	0.427
LVM II	183.30 ± 43.71		200.01 ± 45.09		
Δ [g]	−7.70		6.13		
LVMI I	96.37 ± 16.40	0.117	97.71 ± 19.57	0.191	0.688
LVMI II	92.74 ± 17.62		100.63 ± 20.2		
Δ [g/m^2^]	−3.625		2.92		

LVEDD—left ventricular end-diastolic diameter, LVESD—left ventricular end-systolic diameter, LVESV—left ventricular end-systolic volume, LVEDV—left ventricular end-diastolic volume, LVSV—left ventricular stroke volume, LVEF%—left ventricular ejection fraction, LVM—left ventricular mass, LVMI—left ventricular mass index.

**Table 5 ijerph-19-03326-t005:** Results of blood lipid profile test.

Variable	ST + WBV Group	*p*	ST Group	*p*	Δ ST + WBV vs. Δ Standard
TC I	176.77 ± 38.66	0.586	165.00 ± 32.69	0.156	0.945
TC II	157.77 ± 29.92		157.77 ± 29.92		
Δ [mg/dl]	−7.24		−7.23		
HDL I	52.40 ± 7.89	0.551	50.10 ± 14.40	0.991	0.892
HDL II	53.76 ± 14.01		50.13 ± 13.33		
Δ [mg/dl]	1.35		0.03		
LDL I	99.28 ± 32.58	0.722	90.70 ± 26.58	0.179	0.854
LDL II	101.88 ± 47.77		85.50 ± 25.89		
Δ [mg/dl]	2.6		−5.20		
TG I	119.30 ± 45.27	0.920	121.06 ± 50.25	0.395	0.842
TG II	118.32 ± 44.00		110.80 ± 50.36		
Δ [mg/dl]	−0.978		−10.26		
TC/HDL 1	3.42 ± 0.88	0.918	3.47 ± 0.95	0.0832	0.956
TC/HDL 2	3.44 ± 0.88		3.31 ± 0.94		
Δ [mg/dl]	0.015		−0.165		
LDL/HDL 1	1.93 ± 0.74	0.875	1.93 ± 0.74	0.787	0.873
LDL/HDL 2	1.95 ± 0.77		1.90 ± 0.71		
Δ [mg/dl]	0.021		−0.03		

TC—total cholesterol, HDL—high-density lipoproteins, LDL—low-density lipoproteins, TG—triglycerides.

**Table 6 ijerph-19-03326-t006:** Results of the body mass composition analysis.

Variable	ST+ WBV Group	*p*	ST Group	*p*	Δ ST+ WBV vs. Δ ST
Body mass I	84.01 ± 20.71	0.904	84.81 ± 14.74	0.163	0.875
Body mass II	83.97 ± 20.54		84.30 ± 14.38		
Δ [kg]	−0.035		−0.51		
BMI I BMI II Δ [kg/m^2^]	27.90 ± 4.81 27.89 ± 4.76 −0.012	0.906	28.11 ± 3.78	0.070	0.567
			27.78 ± 3.4		
			−0.33		
FP I	30.85 ± 8.86	0.187	28.56 ± 6.65	0.380	0.865
FP II	29.99 ± 7.95		28.39 ± 7.7		
Δ [%l]	−0.86		−0.17		
FM I	*26.75 ± 12.93*	0.148	24.75 ± 8.89	0.295	0.968
FM II	25.76 ± 11.26		24.57 ± 10.15		
Δ [kg]	−0.99		−0.18		
FFM I	59.26 ± 12.06	0.162	60.06 ± 8.19	0.736	0.925
FFM II	58.82 ± 12.99		59.74 ± 7.69		
Δ [kg]	−0.44		−0.32		
TBW I	44.91 ± 8.82	0.162	43.96 ± 5.99	0.732	0.925
TBW II	44.62 ± 9.51		43.73 ± 5.64		
Δ [kg]	−0.29		−0.23		
FFM-TBW I	14.35 ± 3.24	0.127	16.2 ± 2.2	0.712	0.897
FFM-TBW II	14.20 ± 3.48		16.01 ± 2.05		
Δ [kg]	−0.14		−0.19		
BMR I	1587.12 ± 266.51	0.758	1593.41 ± 193.52	0.177	0.831
BMR II	1584.29 ± 261.7		1590.80 ± 188.25		
Δ [kcal]	−2.83		−2.61		
BMR I	6639.58 ± 1115.25	0.740	6681.35 ± 809.68	0.245	0.856
BMR II	6628.29 ± 1095.54		6665.85 ± 803.42		
Δ [kJ]	−11.29		−15.50		

BMI—body mass index FP—fat percent, FM—fat mass, FFM—fat-free body mass, TBW—total body water FFM-TBW—fat free body mass vs. total body water, BMR—basal metabolic rate.

## Data Availability

Not applicable.

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
