# Peer review of "The Use of Vibration Training in Men after Myocardial Infarction"

_ijerph, 2022, doi:10.3390/ijerph19063326_

Round 1

Reviewer 1 Report

No further comments

Author Response

thank you very much

Reviewer 2 Report

Dear Author 

the errors I previously outlined on table 7 (instead of 3 and this was my tiping error !) are related to disalignment and justification. I guess that it is only a question of margins....

Author Response

Thank you very much

Reviewer 3 Report

Despite improvements made in response to some minor comments, the manuscript requires further revision.

As indicated in the review, the study compares the effect of WBV and resistance training. This is still not adequately outlined in the manuscript. Apart from a short comment added in Material and Methods, no changes were made in the Abstract and Introduction (where the research questions are shown). Similarly, data are consequently presented as WBV vs ST (standard training). If the study had aimed to compare WBV vs ST, WBV should have been added to the standard procedure instead of replacing one of its parts, ie. resistance training.

The advantages of the WBV over the resistance training are not adequately outlined and thus, the rationale of the study remains unclear for the reader. If vibration is risky for patients in NYHA class >I, why do the Authors suggest that NYHA II/III patients would take advantage of it? 

The manuscript should be extensively revised bearing in mind what its rationale was and what strategies were actually compared in the study.

Most importantly, a randomized study should provide high-quality data. The fact that less women met the inclusion criteria [what do the Authors mean by ‘3 (for 10)’?] should not preclude the enrollment. In both versions of the manuscript, sex is not inclusion criterion. To make sure that the study was designed and conducted properly, the Authors should provide the study protocol, NCT number and graphic illustration of the enrollment + randomization process. Otherwise, the quality of the presented data may be understandably put in question.

Also, if only males were included, the Title should be rephrased to ‘…in men after myocardial infarction’.

Author Response

Despite improvements made in response to some minor comments, the manuscript requires further revision.

As indicated in the review, the study compares the effect of WBV and resistance training. This is still not adequately outlined in the manuscript. Apart from a short comment added in Material and Methods, no changes were made in the Abstract and Introduction (where the research questions are shown). Similarly, data are consequently presented as WBV vs ST (standard training). If the study had aimed to compare WBV vs ST, WBV should have been added to the standard procedure instead of replacing one of its parts, ie. resistance training.

We changed title WBV to ST+ WBV to make it more clear.

The advantages of the WBV over the resistance training are not adequately outlined and thus, the rationale of the study remains unclear for the reader. If vibration is risky for patients in NYHA class >I, why do the Authors suggest that NYHA II/III patients would take advantage of it? 

In the Limitation there is a sentence who can be answer to your question: Qualification for the study should include not only patients in NYHA class I, but also patients in NYHA classes II and III due to the lack of differences in the final effect of both forms of training.

The manuscript should be extensively revised bearing in mind what its rationale was and what strategies were actually compared in the study.

Most importantly, a randomized study should provide high-quality data. The fact that less women met the inclusion criteria [what do the Authors mean by ‘3 (for 10)’?] should not preclude the enrollment. In both versions of the manuscript, sex is not inclusion criterion. To make sure that the study was designed and conducted properly, the Authors should provide the study protocol, NCT number and graphic illustration of the enrollment + randomization process. Otherwise, the quality of the presented data may be understandably put in question.

We added in the Matherial and Methods, flow diagram. It shows randomization process.

Also, if only males were included, the Title should be rephrased to ‘…in men after myocardial infarction’. we changed it

Reviewer 4 Report

  1. In line 74 “The study involved 63 males classified as NYHA I.”, but according the Figure 1 there was reported “Randomised (69)” need to clarify.
  2. The authors need to explain “The patients were randomly assigned (random selection) based on two therapeutic strategies: ST group – 29 people; ST+ WBV group – 40 people” in the flow chart; base on which method to randomization?
  3. This study might suggest used the intention-to-treat analysis to include the data of 69 subjects in the analysis to conform to the spirit of RCT and avoid overestimating the effect of intervention.
  4. Data in the tables 2-6 should merged into Table 2 “Basic characteristic of the participants”.

Author Response

Thank you for your comments, there are answers above:

ad.1) We make changes in Figure 1 to make it more clear, why at the begining there were 69 participants and during the research from each group some of patients were excluded.

ad.2) We added the sentence "drawing cards with the name of the group"

ad.3) We would like to refer to this comment but we did not understand your intention. Please, could you explain once again what is your suggestion?

ad.4) We made one table for whole characteristic of both groups.

Reviewer 5 Report

As you said vibrotherapy, as an innovative method of cardiac rehabilitation, can be an excellent alternative, especially for people who are weak and unable to exercise intensively after cardiovascular diseases and your results demonstrated it. New studies are necessary, with a longer observation period, in both sexes and different NYHA classes due to the lack of differences in the final effect of both forms of training. I see vibrotherapy as a potential alternative to usual physical activities in patients with CAD after AMI subjected to PTCA.

I believe the image in Figure 1 should be represented by a man instead of a woman because the study was done only with men.  

Author Response

Thank you for your review. We changed the Figure for unisex.

Round 2

Reviewer 3 Report

According to the flow chart, there seems to be a lot of flaws in the study design and conduction. It is strange that olny 3 women were eligible out of 79 if gender was not an exclusion criterion. It is illicit to exclude patients by gender to avoid 'changing results of statistical analysis' unless the study protocol had been costructed that way. Unfortunately, the Authors do not provide the study protocol. We do not find out what the allocation ratio was and what randomization method was used (>1.5 times more patients were allocated to the experimental arm, ie. 40 vs. 29). Also, according to the diagram, 10 patients were excluded out of the eligible 79, however the numbers given below (7+3+1+2) do not make 10. It is not specified what were 'other reasons'.

Secondly, the Authors do not seem to acknowlegde what the study intervention was. Let me repeat that if resistance training in the standard therapy was replaced with WBV trainig in the experimental arm, these two types of training are actually compared, and not standard vs. WBV.

The advantages of the WBV over the resistance training are not adequately outlined and thus, the rationale of the study remains unclear for the reader. The sentence added in the Limitations section is not answer to this issue.

The changes made in this round of revisions are not sufficient to provide high-quality data.

Author Response

all changes were done

Reviewer 4 Report

OK

This manuscript is a resubmission of an earlier submission. The following is a list of the peer review reports and author responses from that submission.

Round 1

Reviewer 1 Report

Materials and Methods: It seems that NYHA class I is an inclusion criterion but not presented as such in the inclusion criteria group; the authors should comment why the two groups are of different size without any apparent ratio; 

Discussion: There is no need to repeat the numbers already presented in the Tables. This will make the discussion more concise. 

For Tables 1 - 6 there is no referral in the text.

Reviewer 2 Report

The authors presented an original intervention training for males that had a NSTEMI, mainly treated with angioplasty and heart failure classified as stage I NYHA. Patients underwent  reahabilitation program including vibration training with no significance difference in respect to programs recommended by European Society of Cardiology. The study was correctly designed and results, on my opinion, are really interesting.

I must suggest some minor revision including acronimous that not always are specified as well as  errors in table 2  (SP group) and 3 (CPET results). Finally the last sentence of point 4.2 (293-297) is not comprehensible 

Reviewer 3 Report

The authors presented a study on the effects of applied whole body vibration training on exercise tolerance, evaluated through an exercise, blood lipid profile and changes in selected echocardiographic parameters of patients after myocardial infraction. 

Are the 63 patient samples used in the study were on medication or all without any influence of medication? If yes, would the results on vibrotherapy analysis proposed by the authors could also be influenced by type of medication taken by the patient? as taken pills also could influence body metabolism that could effect physical activities..

Reviewer 4 Report

A randomized study comparing the effects of an experimental training program including whole body vibration (WBV) vs standard training program carried out in a group of 63 men after myocardial infarction.

The study shows similar effects of both types of training in terms of exercise capacity and echocardiographic assessment and no effect in blood lipid panel and body composition. Based on the presented data, WBV may seem an interesting component of future cardiac rehabilitation programs. However, the study rationale is not clearly outlined and conclusions are not adequately driven.

In the Abstract, we find out that ‘each session consisted of endurance, general stamina and resistance training. Instead of resistance training, the experimental group performed exercises on vibration platform’. In fact, the study compared the effect of WBV and resistance training, which is not clearly stated throughout the text.

The Authors do not provide uniform information on which advantages of the WBV vs. resistance training were investigated. Throughout the text, various motivations can be found, such as:

  • ‘a good alternative to the sometimes unattractive exercise’ (Introduction)
  • ‘an excellent alternative, especially for people who are weak and unable to exercise intensively’ (Summary)
  • ‘obese patients (…) will benefit’ (Summary)
  • 'patients with circulatory failure in NYHA class II/III (…) will improve their physical activity’ (Summary)

Unfortunately, none of these aspects were addressed in the study - it did not compare patients's interest in the exercise program, 'weak' and obese patients's compliance, and only NYHA I patients were enrolled.

Moreover, the Authors do not acknowledge shortcomings of the WBV, such as the access to the vibration platform. Unlike resistance exercise, this type of training cannot be continued by patients at home which may affect maintaining physical activity.

Minor comments:

  1. the Authors do not clarify why only males were included;
  2. presented data show that only patients after MI (STEMI or NSTEMI) were enrolled; if so, I recommend to rephrase the title adequately (‘The use of whole body vibration training in patients after myocardial infarction’);
  3. age =< 75 is listed as exclusion criterion, however mean age was ca. 53/55 years; I suppose it is a typo;
  4. PCI should be replaced with PTCA in Table 5.